# Categorical and phenotypic image synthetic learning as an alternative to federated learning

Nghi C. D. Truong [1] ✉, Chandan Ganesh Bangalore Yogananda[1], Benjamin C. Wagner [1], James M. Holcomb[1], Divya D. Reddy[1], Niloufar Saadat[1], Jason Bowerman[1], Kimmo J. Hatanpaa[2], Toral R. Patel [3], Baowei Fei[1,4], Matthew D. Lee [5], Rajan Jain [5,6], Richard J. Bruce[7], Ananth J. Madhuranthakam[8], Marco C. Pinho[1] & Joseph A. Maldjian [1] ✉

Multi-center collaborations are crucial in developing robust and generalizable machine learning models in medical imaging. Traditional methods, such as centralized data sharing or federated learning (FL), face challenges, including privacy issues, communication burdens, and synchronization complexities. We present CATegorical and PHenotypic Image SyntHetic learnING (CATphishing), an alternative to FL using Latent Diffusion Models (LDM) to generate synthetic multi-contrast three-dimensional magnetic resonance imaging data for downstream tasks, eliminating the need for raw data sharing or iterative inter-site communication. Each institution trains an LDM to capture site-specific data distributions, producing synthetic samples aggregated at a central server. We evaluate CATphishing using data from 2491 patients across seven institutions for isocitrate dehydrogenase mutation classification and three-class tumor-type classification. CATphishing achieves accuracy comparable to centralized training and FL, with synthetic data exhibiting high fidelity. This method addresses privacy, scalability, and communication challenges, offering a promising alternative for collaborative artificial intelligence development in medical imaging.

Collaborative multi-center studies are essential for developing robust and generalizable machine-learning models in medical imaging. Such studies typically rely on pooling diverse datasets to account for variability in imaging protocols, patient populations, disease characteristics, and clinical practices. This diversity enhances the models' generalizability and robustness, enabling them to perform reliably across different clinical settings. However, sharing patient data between institutions introduces significant challenges, as institutions must maintain patient privacy and comply with data protection regulations[1]. Federated learning (FL)[2–4] has emerged as a promising solution to address privacy concerns in collaborative learning. In FL, institutions train models locally on their data, sharing only model parameters and thereby mitigating the need to exchange raw patient data. However, FL still faces challenges, including communication overhead and the complexity of model synchronization across centers.

[1]Department of Radiology, University of Texas Southwestern Medical Center, Dallas, TX, USA. [2]Department of Pathology, University of Texas Southwestern Medical Center, Dallas, TX, USA. [3]Department of Neurological Surgery, University of Texas Southwestern Medical Center, Dallas, TX, USA. [4]Department of Bioengineering, University of Texas at Dallas, Richardson, TX, USA. [5]Department of Radiology, NYU Grossman School of Medicine, New York, NY, USA. [6]Department of Neurosurgery, NYU Grossman School of Medicine, New York, NY, USA. [7]Department of Radiology, University of Wisconsin-Madison, Madison, WI, USA. [8]Department of Radiology, Mayo Clinic, Rochester, MN, USA. ✉e-mail: nghicd.truong@gmail.com; Joseph.Maldjian@UTSouthwestern.edu

Recent advancements in generative AI[5,6], particularly diffusion models (DMs)[6–9], offer new avenues to address these limitations. In this context, we introduce CATegorical and Phenotypic Image SyntHetic learnING (CATphishing) as an alternative to FL for medical imaging applications. CATphishing leverages recent advancements in generative AI and diffusion models to create a scalable and privacy-preserving framework for collaborative multi-center studies. In CATphishing, latent diffusion models (LDMs)[6,10] are trained locally at each participating site to capture the underlying data distribution of the original data. These models generate synthetic, "site-like" samples that are aggregated at a central server for downstream tasks such as image classification. By eliminating the need for direct data sharing or complex multi-site communication, CATphishing offers a scalable, privacy-preserving solution for collaborative studies.

Recent studies have extensively explored the application of DMs in various medical imaging contexts, particularly in generating 3D medical images such as MRI scans[10–14], demonstrating their capacity to model complex 3D imaging distributions. Although some works have integrated DMs into FL frameworks to enhance data diversity and address heterogeneity, these have primarily focused on 2D color images. For instance, Zhao et al.[15] proposed a data augmentation-based FL method where a diffusion model was trained via FL to alleviate non-independent and identically distributed (non-IID) data issues. Mendieta et al.[16] employed a one-shot FL approach to train diffusion models: models were trained locally and then gathered at a central server to generate training data for a global diffusion model. Other studies have also explored various FL strategies for training diffusion models[17–19], but none have addressed applications involving 3D MRI as a standalone alternative to FL. While previous studies have used generative models, including DMs, to enhance tasks such as segmentation[14,20,21] or mutation prediction[22], our CATphishing framework uniquely positions synthetic data generation via LDM as a communication-free, privacy-preserving alternative to FL in collaborative multi-center medical imaging studies involving 3D MRI.

To evaluate the effectiveness of CATphishing, we implemented two clinically significant classification tasks: IDH mutation classification and tumor-type classification based on the WHO 2021 criteria[23]. We assembled a large and diverse dataset from multiple sources, consisting of 2491 real patient cases, carefully divided into distinct training and testing cohorts to simulate a realistic multi-site research scenario. For the IDH mutation classification task, separate sites were used for training and independent testing to assess generalizability across diverse populations and imaging protocols. For the tumor-type classification task, we employed a two-stage pipeline: Stage 1 distinguishes IDH-mutated and wildtype, and Stage 2 further classifies IDH-mutated cases into oligodendroglioma or astrocytoma, again using different sites for training and testing to simulate federated or collaborative scenarios. LDMs were trained locally at each site to generate synthetic multi-contrast MRI data representative of their respective datasets' distribution. These synthetic datasets were then aggregated centrally to train an IDH classification model.

Here, we demonstrate that models trained solely on synthetic data generated by LDMs achieve comparable performance to those trained on real, shared data or through FL, in both classification tasks. These results underscore the generalizability and scalability of the CATphishing framework and highlight its potential to support diverse medical imaging applications, such as segmentation, detection, and multi-class classification, while addressing privacy concerns and enabling secure, multi-institutional collaboration.

## Results
### Datasets
This study utilized retrospective MRI scans from four publicly available and three internal datasets. The public datasets included The Cancer Genome Atlas (TCGA)[24], the Erasmus Glioma Database (EGD)[25], the University of California San Francisco Preoperative Diffuse Glioma MRI dataset (UCSF)[26], and the University of Pennsylvania glioblastoma (UPenn) cohort[27]. Internal datasets were collected from three institutions: UT Southwestern Medical Center (UTSW for the first cohort and UTSWp2 for the second distinct cohort), New York University (NYU), and the University of Wisconsin-Madison (UWM). This diverse data collection ensures the representation of a wide range of patient populations across centers, which is crucial for evaluating model robustness.

Data included in this study were required to have preoperative MRI scans of four sequences: T1-weighted (T1), post-contrast T1-weighted (T1C), T2-weighted (T2), and T2-weighted fluid-attenuated inversion recovery (FLAIR). In total, the dataset consisted of 2491 unique patients divided into completely independent training and testing cohorts. For the IDH classification task, the training cohort included 1014 patients from TCGA ($n = 198$), UTSW ($n = 360$), and EGD ($n = 456$), which were used to train both the IDH classification models and LDMs. The testing cohort comprised 1477 patients from NYU ($n = 181$), UWM ($n = 218$), UCSF ($n = 495$), UPenn ($n = 419$), and UTSWp2 ($n = 164$). For the tumor-type classification experiment, the training cohort included 691 patients from TCGA, UTSW, and NYU. The testing cohort comprised 1294 patients from EGD, UWM, UCSF, and UTSWp2. A detailed summary of patient characteristics, IDH mutation, and tumor-type statuses across the datasets is provided in Table 1.

### Image preprocessing
Except for the UCSF dataset, all datasets were preprocessed using the federated tumor segmentation (FeTS) tool[28]. The pre-processing pipeline included co-registering MRI scans to a template atlas, correcting for bias field distortions, and skull stripping. For the TCGA, UTSW, NYU, and EGD datasets, the FeTS tool was also used to segment the whole tumor. The resulting tumor masks served as inputs for training the LDMs and as targets while training the classification models using real data. For the UCSF dataset, we used pre-processed skull-stripped images provided by the original data source, as access to the raw data was unavailable.

Following FeTS pre-processing, skull-stripped images were cropped to the bounding box of the non-zero area, and z-score normalization was applied to all non-zero voxels for each sequence. These z-score normalization images were used to train both the classification models and the LDMs.

### Overview of the study
In this study, we introduce CATphishing as an alternative to FL for multi-site collaborations. Traditionally, FL allows for collaborative model training across multiple sites without sharing raw patient data. In the standard federated approach (Fig. 1(a)), each participating center trains the classification model locally and then shares only the model weights with a central server. The central server aggregates these local models into a global model, which is then redistributed back to the centers for the next round of training. This iterative process ensures that no raw data is shared between centers, maintaining patient privacy while building a robust classification model.

Our proposed CATphishing method leverages LDMs to generate synthetic MR images, offering an alternative for multi-site collaboration (Fig. 1b). Specifically, each participating center independently trains an LDM on its local dataset to capture the underlying data distribution. These trained LDMs are then sent to a central server, where they are used to generate synthetic MRI samples that represent the data from each center. By aggregating synthetic samples from all centers, we create a comprehensive synthetic dataset that can be used to train a centralized downstream task, including the IDH classification model or a two-stage tumor-type classification pipeline.

**Table 1 | Patient characteristics of different datasets**

| | TCGA (n = 198) | UTSW (n = 360) | EGD (n = 456) | NYU (n = 181) | UWM (n = 218) | UCSF (n = 495) | UPenn (n = 419) | UTSWp2 (n = 164) |
|---|---|---|---|---|---|---|---|---|
| *IDH status* | | | | | | | | |
| Mutated | 91 | 104 | 150 | 48 | 18 | 103 | 11 | 50 |
| Wildtype | 107 | 256 | 306 | 133 | 200 | 392 | 408 | 114 |
| *Tumor type* | | | | | | | | |
| Oligodendroglioma | 27 | 40 | 71 | 21 | 2 | 9 | N/A | 16 |
| Astrocytoma | 63 | 29 | 74 | 20 | 13 | 84 | N/A | 17 |
| Glioblastoma | 107 | 256 | 306 | 128 | 198 | 392 | N/A | 112 |
| *Sex* | | | | | | | | |
| Male | 102 | 195 | 284 | 95 | 128 | 296 | 254 | 95 |
| Female | 95 | 147 | 171 | 86 | 90 | 199 | 165 | 69 |
| Unknown | 1 | 18 | 1 | 0 | 0 | 0 | 0 | 0 |
| *Age* | | | | | | | | |
| >65 | 41 | 91 | 136 | 50 | 86 | 153 | 176 | 58 |
| <=65 | 156 | 248 | 280 | 131 | 132 | 342 | 243 | 106 |
| Unknown | 1 | 21 | 40 | 0 | 0 | 0 | 0 | 0 |
| Mean | 51.9 | 53.8 | 56.0 | 55.2 | 59.8 | 56.9 | 62.7 | 55.5 |
| STD | 15.3 | 15.4 | 15.1 | 15.9 | 13.8 | 15.1 | 11.9 | 16.6 |

Data are presented as the number of patients.

*TCGA* The Cancer Genome Atlas, *UCSF* University of California San Francisco Preoperative Diffuse Glioma MRI dataset, *EGD* Erasmus Glioma Database, *UPenn* The University of Pennsylvania glioblastoma cohort, *UTSW* UT Southwestern Medical Center, *NYU* New York University, *UWM* the University of Wisconsin-Madison, *UPenn* University of Pennsylvania glioblastoma cohort, *UTSWp2* UT Southwestern Medical Center part 2, *STD* standard deviation.

## Evaluation of the synthetic MRI samples

Figure 2 shows representative axial slices from four multi-contrast MRI samples generated by the LDMs trained on the TCGA, UTSW, and EGD datasets. The LDMs also generated the tumor masks (red contour) alongside the 4 MRI sequences. The LDMs successfully produced realistic brain anatomy and expected tumor characteristics associated with IDH mutation status. To further illustrate the diversity of tumor imaging phenotypes captured by the LDMs, we have included additional synthetic examples in Supplementary Figs. S2–S4 covering three tumor types: oligodendroglioma (IDH-mutated, 1p19q co-deleted), astrocytoma (IDH-mutated, 1p19q non-codeleted), and glioblastoma (IDH-wildtype). These Supplementary figures highlight variations in tumor location, size, and enhancement patterns, supporting the LDMs' capacity to model biologically meaningful heterogeneity.

To assess the realism of the synthetic images, we employed the Fréchet Inception Distance (FID)[29], a widely used metric to estimate the distances between feature representations of synthetic and real images. A smaller FID indicates greater similarity in the distribution between the real and generated images, reflecting higher synthetic image fidelity. FID was calculated for each MRI sequence using 300 synthetic and 300 real samples. Prior to calculating the FID scores, both synthetic and real samples were preprocessed by zeroing the background and applying z-score normalization to the brain area.

Table 2 presents the FID scores between synthetic and real MRI samples across various datasets (TCGA, UTSW, and EGD) and MRI sequences (T1, T1C, T2, and FLAIR). Synthetic samples generated by models trained on a specific dataset generally show low FID scores when compared to real samples from the same dataset, indicating that the LDMs effectively learn dataset-specific distributions. Notably, synthetic UTSW and EGD data exhibit lower FID scores with their respective real datasets than synthetic TCGA data does with its real dataset, suggesting higher fidelity in the UTSW and EGD cases.

Cross-dataset comparisons consistently yield higher FID scores, reflecting the distinct distributions among the datasets. For instance, synthetic UTSW and synthetic EGD exhibit notably high FID scores compared to datasets they were not trained on. Similar trends are observed for synthetic TCGA data in cross-dataset comparisons,

although the FID scores were slightly lower than those for synthetic UTSW and EGD. This suggests that synthetic TCGA data is more similar to the other datasets than synthetic UTSW and EGD data. Overall, all synthetic data achieved relatively strong fidelity with their real counterparts, while higher FID scores in cross-dataset comparisons highlight the dataset-specific nature of the generated images. These results underscore the ability of the LDMs to generate synthetic data that closely resembles their training datasets.

We further assessed the quality of the synthetic MRI images using no-reference image quality assessment metrics: the Blind/Reference-less Image Spatial Quality Evaluator (Brisque)[30,31] and the Perception-based Image Quality Evaluator (PIQE)[32]. We computed these metrics on 300 synthetic and 300 real samples for each dataset and MRI sequence. Table 3 presents a quantitative comparison of the image quality metrics between real and synthetic MRI data. The Brisque scores for synthetic images were consistently lower than those of their real counterparts across all datasets and modalities, indicating that the synthetic images contained less noise and fewer artifacts as detected by this metric.

In contrast, the PIQE metric yielded mixed results for the synthetic images. While they outperformed real images in certain modalities within the EGD dataset, real images achieved better PIQE scores in most cases. These results suggest that the synthetic images have relatively lower perceptual quality compared to their real counterparts. This indicates that although the synthetic images may exhibit fewer pixel-level noise and artifacts, as reflected by the lower Brisque scores, their higher-level structural fidelity requires further improvement to match the perceptual quality of real images.

## Performance of IDH classification models

To evaluate the effectiveness of CATphishing, we conducted a comprehensive comparison against traditional training strategies. Specifically, we benchmarked CATphishing against (1) centralized training using real shared data from three sites and (2) FL using real data from the same sites without data sharing. For each training strategy, we trained IDH classification models and evaluated their performance using key metrics, including accuracy (ACC), sensitivity

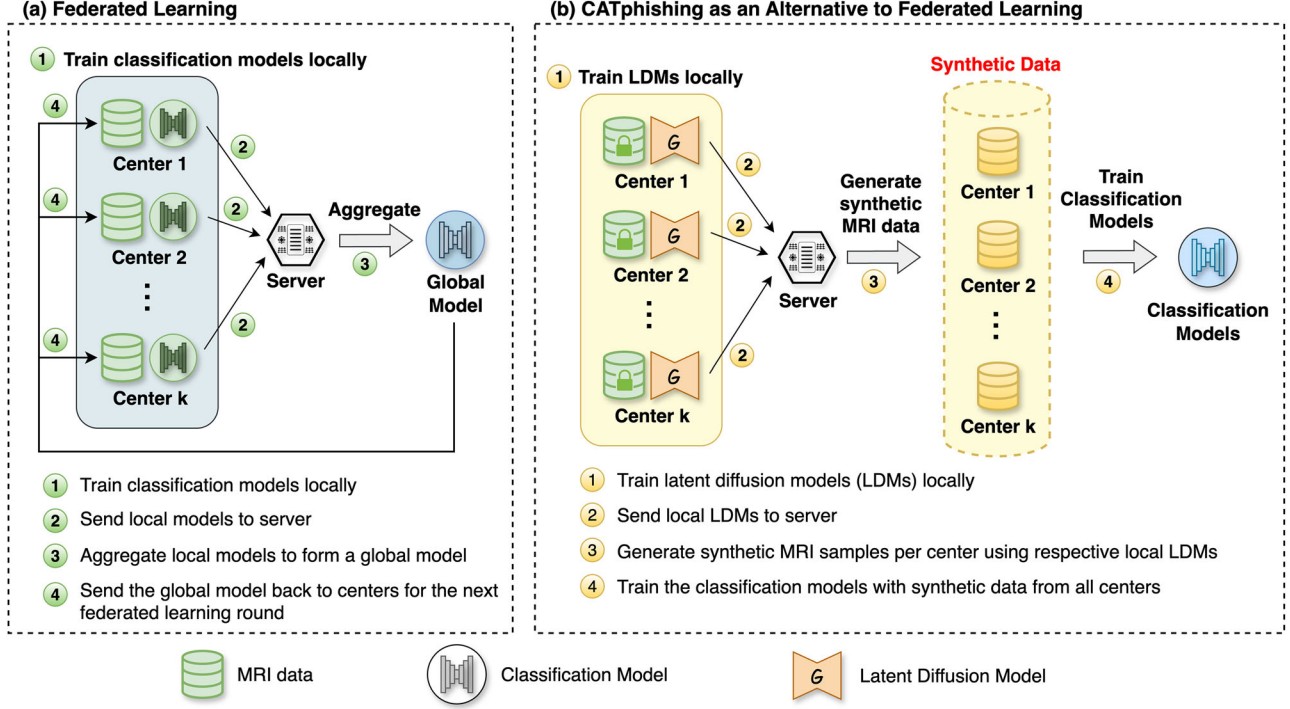

**Fig. 1 | Comparison of Federated Learning and CATphishing workflows.**
**a** Federated Learning trains local models on real data at each center, then aggregates model updates centrally without sharing raw data. **b** CATphishing trains generative models locally to create synthetic datasets, which are combined and used to train a final centralized machine learning model.

(SEN), specificity (SPE), and receiver operating characteristic area under the curve (AUC), on an independent test set from five sites: UCSF, NYU, UWM, UPenn, UTSW part 2. This evaluation allowed us to determine whether models trained solely on synthetic data can achieve performance comparable to those trained on real data in both centralized and federated settings. Centralized training serves as the reference point, while FL and CATphishing were compared against it using McNemar's test.

Table 4 presents the classification performance of different models across various testing datasets. The ROC curves for the combined test sets, including UCSF, NYU, UWM, UPenn, and UTSWp2, with the AUC values and their corresponding 95% CIs are depicted in Fig. 3. Centralized training with real data achieved a strong overall accuracy of 96.2%, sensitivity of 86.5%, specificity of 98.0%, and an AUC of 0.979, underscoring its robustness when data from all sites were combined into a single shared dataset. FL using the FedAvg algorithm with 100 rounds demonstrated highly comparable performance, with a slightly lower overall ACC of 95.8%. This suggests that FL, even without direct data sharing, can approximate the predictive accuracy of centralized approaches while preserving data privacy across sites. McNemar's test indicates no statistically significant difference in classification compared to centralized training.

Centralized training using synthetic data generated by LDMs through the CATphishing framework showed excellent and comparable results, achieving an overall accuracy of 95.5%, sensitivity of 86.1%, specificity of 97.2%, and an AUC of 0.966. While slightly lower than the results from FL and centralized training with real data, these metrics highlight the potential of synthetic data to enable high-quality model training. McNemar's test demonstrated no statistically significant difference in predictions compared to centralized training with real data, further validating the utility of the CATphishing approach for preserving subject anonymity and local data integrity while maintaining robust classification performance.

## Performance of two-stage tumor-type classification models
To further evaluate the effectiveness of CATphishing, we applied it to a multi-class tumor-type classification task based on the WHO 2021 criteria. This task was implemented as a two-stage pipeline. The first stage distinguishes IDH-wildtype glioblastomas from IDH-mutated astrocytomas and oligodendrogliomas. The second stage further classifies IDH-mutated cases into oligodendroglioma and astrocytoma. While the first stage mirrors our earlier IDH experiment, this experiment adopted a different training and testing split. The additional IDH classification results evaluated on all IDH cases under this new split are provided in Supplementary Note 6. Table 5 presents the performance solely for patients with available tumor-type status.

Table 5 summarizes the performance of each stage and the final three-class tumor-type classification across centralized training, FL, and CATphishing. In stage 1, all three approaches achieved similarly high performance, with no statistically significant differences between FL or CATphishing and the centralized training baseline. These results further validate the robustness and reliability of IDH classification across different collaborative learning strategies.

Stage 2, which classifies oligodendroglioma versus astrocytoma within IDH-mutated cases, remained more challenging due to the subtle imaging differences between the two subtypes. Centralized training achieved an accuracy of 76.2% (AUC = 0.844), while FL and CATphishing achieved 75.2% accuracy, with AUCs of 0.839 and 0.821, respectively. When evaluating the overall performance across both stages, the final three-class tumor-type classification accuracies were also comparable: 91.9% for centralized training, 91.5% for FL, and 90.9% for CATphishing. Confusion matrices (Fig. 4) revealed similar misclassification patterns across all methods, with the most common errors occurring between oligodendroglioma and astrocytoma.

Statistical comparisons using the McNemar test for stage 1 and stage 2 and the Stuart-Maxwell test for the final three-class

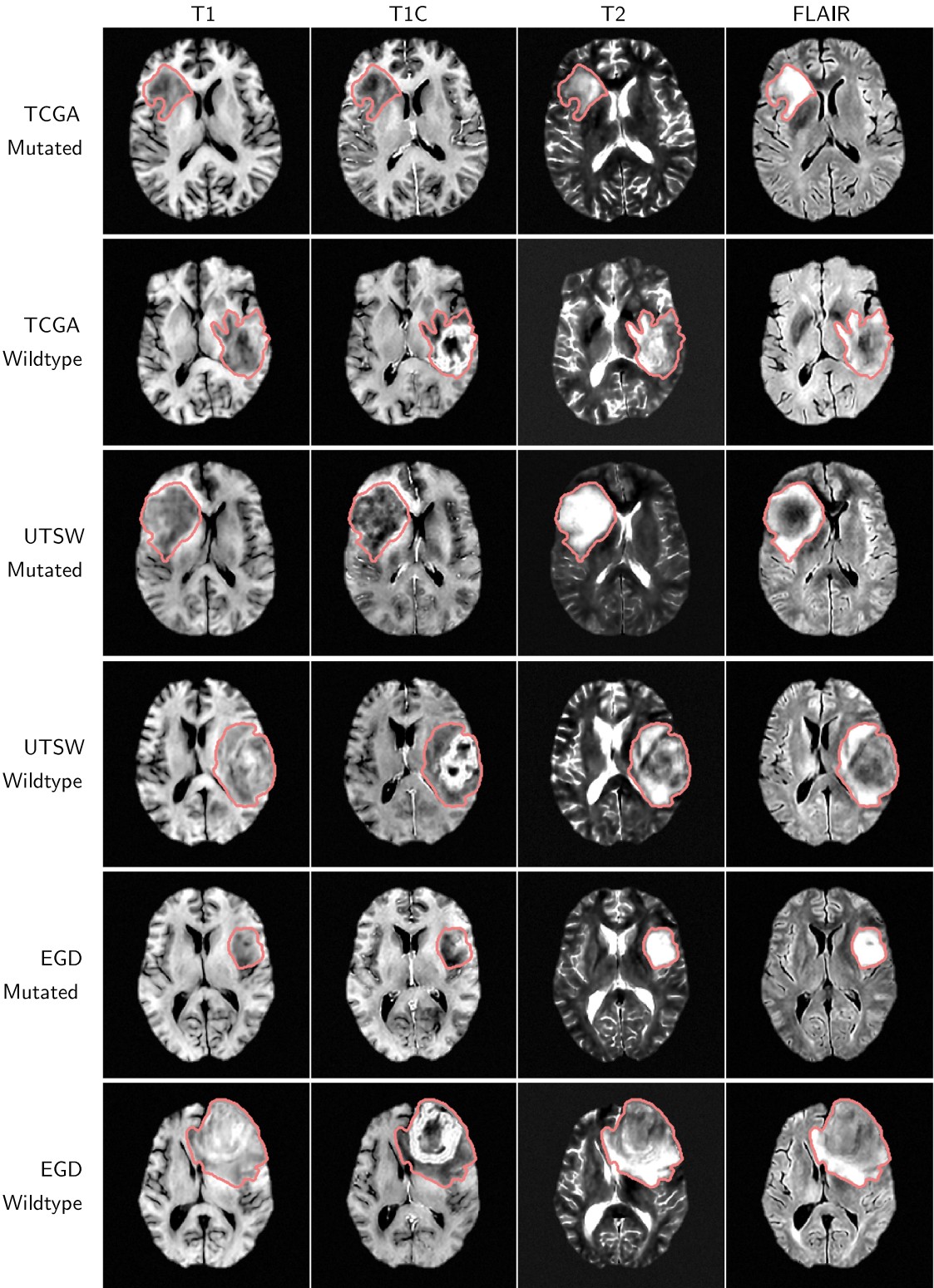

**Fig. 2 | Multi-contrast MRI samples generated by the LDM with the conditional IDH mutation status for the TCGA, UTSW, and EGD sites.** The whole tumor masks (red contour) are also part of the generated output.

classification confirmed no significant differences between CATphishing and centralized training. These findings confirm that CATphishing-generated synthetic data can effectively support complex, multi-class tumor classification tasks, offering a viable, privacy-preserving alternative to traditional data-sharing or FL approaches.

## Discussion

This study introduces CATphishing as an alternative approach to FL using LDMs. In this method, LDMs are trained locally at each participating site to capture the underlying data distribution of the original data. Site-specific synthetic samples are then generated and aggregated at a central server for a downstream task. This approach offers

**Table 2 | Evaluation of the fidelity of the synthetic brain tumor MRI samples compared to the real data**

|  | T1 | | | T1C | | | T2 | | | FLAIR | | |
|---|---|---|---|---|---|---|---|---|---|---|---|---|
|  | Real TCGA | Real UTSW | Real EGD | Real TCGA | Real UTSW | Real EGD | Real TCGA | Real UTSW | Real EGD | Real TCGA | Real UTSW | Real EGD |
| Synthetic TCGA | 65.6 | 74.9 | 97.2 | 87.8 | 103.1 | 127.1 | 67.4 | 71.2 | 82.3 | 70.1 | 81.0 | 87.9 |
| Synthetic UTSW | 290.1 | 56.9 | 287.7 | 304.0 | 66.9 | 304.2 | 226.6 | 35.3 | 237.1 | 273.2 | 47.8 | 264.8 |
| Synthetic EGD | 290.3 | 301.6 | 45.1 | 302.9 | 307.8 | 59.3 | 236.6 | 243.7 | 39.0 | 267.4 | 270.7 | 38.8 |

FID scores were calculated for each MRI sequence using a set of 300 synthetic and 300 real samples.

several advantages, particularly in settings where data privacy, communication overhead, and data-sharing restrictions are major concerns. Unlike traditional centralized training methods that require merging datasets from multiple institutions or FL approaches involving distributed data with ongoing communication, CATphishing can sidestep these complexities.

In FL, model parameters must be exchanged iteratively between institutions or nodes. Although less privacy-invasive than sharing raw data, this communication still introduces logistical challenges, potential security risks, and significant overhead due to the continuous back-and-forth exchanges at all sites[33–35]. Although asynchronous FL offers more flexibility by allowing independent uploads and reducing synchronization burdens, it introduces other complexities such as handling stale updates, version control, and fairness issues. CATphishing avoids these issues altogether by allowing each participating site to independently train models locally without the need for inter-site communication. This independence simplifies collaboration, reduces latency, and enhances data security and robustness against network disruptions.

Additionally, CATphishing offers enhanced scalability. Institutions can independently train their models without requiring alignment or coordination with other sites. This makes it easier to include or exclude sites from collaboration, as it does not require altering communication protocols or agreements. New partners can easily adopt the LDMs for synthetic sample generation without complex setup processes, facilitating broader participation in multi-institutional studies.

Evaluating the fidelity of the synthetic data, the FID scores suggested that the synthetic data was similar to its training dataset and distinct from others. In other words, LDMs can generate site-representative samples. Thus, models trained with synthetic data maintain the advantages of enriched data diversity without the risks or legal implications of cross-institutional data sharing. By ensuring that each synthetic dataset maintains fidelity to its respective real counterpart, classification models benefit from a more comprehensive training set, thereby potentially improving their robustness and adaptability when tested on data from diverse sources.

One critical concern in multi-site studies and generative modeling is the risk of membership inference attacks (MIA)[36–38]. CATphishing minimizes these risks by confining model training strictly within each institution. While FL reduces privacy concerns by sharing model parameters rather than raw data, the iterative nature of FL communication can still create windows of vulnerability. In contrast, CATphishing involves a one-time sharing of LDMs or synthetic data, thereby reducing the temporal risks associated with training dynamics. To directly assess the risk of membership leakage from the synthetic data, we conducted an SSIM-based MIA via a distance-based method to compare real training ("member") and hold-out ("non-member") samples (Supplementary Note 4). The results showed chance-level performance (AUC = 0.493) and highly overlapping SSIM distributions (Supplementary Fig. S1), indicating that the synthetic data does not reveal identifiable training information. In addition, CATphishing applied several privacy-preserving strategies, including the use of

preprocessed skull-stripped images in NifTI format to remove identifiable anatomical features, applying data augmentation during LDM training, incorporating drop-out layers, and employing gradient clipping to minimize overfitting and reduce MIA risk[39,40]. Future work should include rigorous assessments to evaluate the resilience of LDMs and their synthetic data against MIAs, further strengthening the privacy-preserving claims of the CATphishing approach.

CATphishing exhibited slight performance limitations compared to centralized training on real data. While the synthetic data closely mimicked the distribution of real data within the same dataset, it might not fully encapsulate the tumor features in our case study. These results emphasize the need for further refinement of the LDM to ensure a comprehensive representation of subtle features critical for accurate IDH classification. Moreover, the quality of the synthetic image still needs improvement, especially the higher-level structure of the whole brain. Enhancing the diversity of the generated samples is also crucial to improve the performance of the IDH classification model in our case study, as well as the performance of high-level applications in general.

While the study primarily focused on IDH mutation classification and three-class tumor-type classification using routinely acquired structural MRI sequences (T1, T1C, T2, and FLAIR), the broader applicability of CATphishing to other clinical tasks and imaging modalities remains a promising direction. Structural MRI was chosen for its widespread availability, consistency across institutions, and foundational role in clinical neuro-oncology workflows. However, given the modality-agnostic design of LDMs, CATphishing could accommodate advanced functional (e.g., diffusion tensor imaging, resting-state fMRI) and metabolic MRI modalities (e.g., arterial spin labeling, chemical exchange saturation transfer), provided that sufficiently high-quality training data are available. Broadening the application of CATphishing to diverse imaging and diagnostic scenarios could unlock new possibilities for multi-center collaborations in healthcare. Overall, this study lays the groundwork for synthetic data generation in medical imaging and highlights the potential of diffusion models as a scalable tool for collaborative AI development in healthcare.

## Methods

This study was conducted in full compliance with all relevant ethical regulations. The study protocol was reviewed and approved by the Institutional Review Board (IRB) at UT Southwestern Medical Center. Given the retrospective nature of the research and the use of fully anonymized internal data as well as publicly available datasets, the IRB granted a waiver of informed consent in accordance with institutional and federal guidelines. All internal data were de-identified before analysis, and the study was conducted in compliance with the Health Insurance Portability and Accountability Act.

In the study, sex information for each dataset was obtained by self-reporting as documented by the respective data sources. This information was recorded only for descriptive purposes and was not considered in the study design, statistical analyses, or outcome evaluation. The classification models were developed and evaluated without incorporating sex and other demographic variables, and sex

information was not used during the training of the generative models. No sex- or gender-based analyses were conducted because the study's primary focus was on technical model development and evaluation rather than on investigating sex- or gender-related impacts.

## Classification model

Both the IDH classification and the three-class tumor-type classification models leverage the semantic segmentation principle, in which each voxel within the whole tumor is assigned a class label (IDH-mutated vs. wildtype or oligodendroglioma vs. astrocytoma). For training, the tumor volume was labeled according to its corresponding ground-truth class, enabling the models to learn from the spatial distribution of imaging features across the tumor region. This voxel-level classification strategy allows the model to capture spatial heterogeneity within tumors, providing a detailed map of mutation probabilities that enhances the accuracy of the overall classification.

We developed an enhanced version of the traditional UNet architecture[41] by employing residual blocks[42] in both the encoder and decoder paths. Residual blocks facilitate the flow of information from earlier to deeper layers, allowing the network to learn more complex and abstract features. Additionally, the Mamba module[43,44] was integrated into the encoder branch, further improving the model's ability to capture critical features relevant to the tumor classification task.

To ensure robustness and generalizability, we employed a fivefold cross-validation training strategy[41]. All classification models were trained for 1000 epochs using the stochastic gradient descent optimizer. The results of these five folds were then ensembled to produce the final classification. We averaged the predicted probabilities across all tumor voxels for subject-wise classification, assigning the class with the higher average probability as the subject-wise final classification. Details of network architecture and training hyperparameters are provided in Supplementary Note 1.

## Federated learning

In the FL approach, each center trains the IDH classification model locally using its own MRI data. As mentioned in the Datasets section, we simulated a multi-center study using 3 datasets: TCGA, UTSW, and EGD for the IDH classification task and TCGA, UTSW, and NYU for the tumor-type classification. The classification models were trained locally using these three datasets. We developed a federated implementation based on our training framework and used the federated averaging (FedAvg) method[4] to aggregate the models' weights from all centers. FL was conducted over 100 rounds. After each round, the server aggregated the weights of three local models to form a global model, which was subsequently distributed back to the centers for the next training rounds. Federated training employed the same preprocessed data and training hyperparameters as those used in centralized training, including the number of epochs, optimizer settings, and learning rates. This ensured consistency between the federated and centralized models. Additional details about the FL implementation can be found in Supplementary Note 2.

## Latent Diffusion Model (LDM)

We have developed an advanced LDM designed to generate synthetic multi-contrast 3D MRI data. Our model extends the foundational architecture proposed by Rombach et al.[6], originally designed for 2D images, into a 3D multi-contrast framework suitable for volumetric MRI data. Unlike our previous work[10], where we provided

**Table 3 | Comparison of image quality between real and synthetic samples using no-reference image quality assessment metrics: Blind/Referenceless Image Spatial Quality Evaluator (Brisque) and Perception-based Image Quality Evaluator (PIQE)**

| | Brisque ↓ | | | | PIQE ↓ | | | |
|---|---|---|---|---|---|---|---|---|
| | T1 | T1C | T2 | FLAIR | T1 | T1C | T2 | FLAIR |
| Real TCGA | 58.7 | 56.5 | 57.7 | 59.7 | *41.6* | *40.5* | *42.9* | *42.8* |
| Synthetic TCGA | *52.6* | *53.7* | *51.3* | *53.2* | 48.9 | 46.3 | 47.8 | 46.4 |
| Real UTSW | 59.6 | 57.8 | 57.1 | 60.2 | *42.9* | *41.4* | *43.1* | *42.8* |
| Synthetic UTSW | *50.3* | *52.0* | *48.2* | *48.4* | 47.3 | 44.0 | 46.1 | 44.3 |
| Real EGD | 59.1 | 57.5 | 56.4 | 58.1 | 48.2 | 43.9 | 46.0 | *41.4* |
| Synthetic EGD | *45.4* | *44.3* | *42.5* | *43.0* | *46.9* | *42.1* | *44.9* | 42.5 |

Source data are provided as a Source Data file.

Italicized values in Table 3 indicate the lower Brisque or PIQE scores when comparing real and synthetic images for each modality and dataset.

**Table 4 | Summary of the IDH mutation status classification performance of different models trained on real or synthetic MRI data**

| Experiments | Training data | Metrics | UCSF | NYU | UWM | UPenn | UTSWp2 | Overall | McNemar Test |
|---|---|---|---|---|---|---|---|---|---|
| Centralized Training | Real TCGA + Real UTSW + Real EGD | ACC | 96.2 | 94.5 | 95.9 | 97.9 | 94.5 | 96.2 | - |
| | | SEN (MT) | 88.4 | 87.5 | 77.8 | 72.7 | 88.0 | 86.5 | |
| | | SPE (WT) | 98.2 | 97.0 | 97.5 | 98.5 | 97.4 | 98.0 | |
| | | AUC | 0.980 | 0.966 | 0.969 | 0.978 | 0.992 | 0.979 | |
| FL using the FedAvg algorithm with 100 FL rounds | Real TCGA + Real UTSW + Real EGD | ACC | 95.6 | 92.8 | 95.4 | 97.9 | 95.1 | 95.8 | $\chi^2(1) = 2.45$, $p = 0.1175$ |
| | | SEN (MT) | 89.3 | 85.4 | 77.8 | 63.6 | 92.0 | 87.0 | |
| | | SPE (WT) | 97.2 | 95.5 | 97.0 | 98.8 | 96.5 | 97.4 | |
| | | AUC | 0.983 | 0.968 | 0.961 | 0.975 | 0.988 | 0.980 | |
| Centralized Training using Synthetic Samples | Synthetic TCGA + Synthetic UTSW + Synthetic EGD | ACC | 94.8 | 92.8 | 95.0 | 98.1 | 94.5 | 95.5 | $\chi^2(1) = 1.31$, $p = 0.2531$ |
| | | SEN (MT) | 83.5 | 91.7 | 88.9 | 72.7 | 88.0 | 86.1 | |
| | | SPE (WT) | 97.7 | 93.2 | 95.5 | 98.8 | 97.4 | 97.2 | |
| | | AUC | 0.972 | 0.952 | 0.962 | 0.960 | 0.962 | 0.966 | |

Statistical comparisons of classification results among methods are performed using the McNemar test. All tests are two-sided, and no adjustments are applied

Sensitivity (SEN) and specificity (SPE) correspond to the accuracy of the mutated (MT) and wild-type (WT) classes, respectively.

*ACC* accuracy, *AUC* area under the receiver operating characteristic curve, *EGD* Erasmus Glioma Database, *NYU* New York University, *TCGA* The Cancer Genome Atlas, *UCSF* University of California San Francisco Preoperative Diffuse Glioma MRI dataset, *UPenn* University of Pennsylvania glioblastoma cohort, *UTSW* University of Texas Southwestern Medical Center, *UWM* University of Wisconsin–Madison, *UTSWp2* University of Texas Southwestern Medical Center part 2.

Source data are provided as a Source Data file.

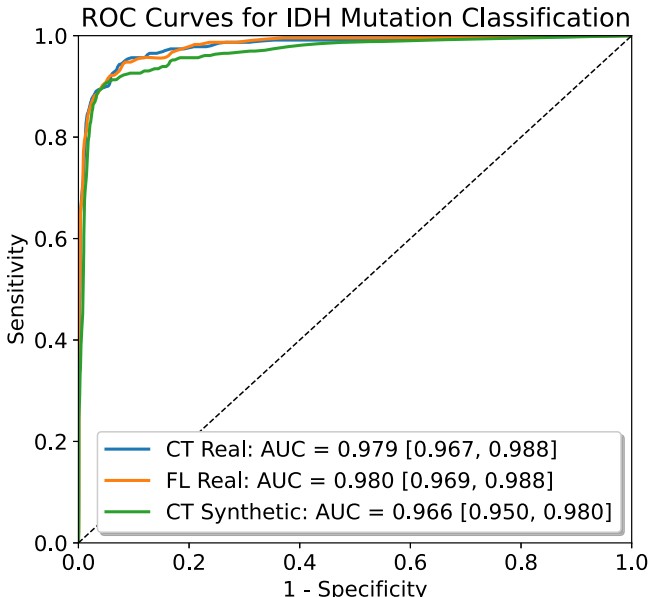

**Fig. 3** | Receiver operating characteristic (ROC) curves of the combined test sets for the models trained using (1) centralized real data (CT Real), (2) FL real data (FL Real), and (3) centralized synthetic data (CT Synthetic). AUC area under the receiver operating characteristic curve. Source data are provided as a Source Data file.

the whole tumor mask and text conditions to generate synthetic mutation-specific brain tumor MRI samples, our LDM is tailored for multi-site studies by generating the whole tumor mask alongside the multi-contrast MR images. This approach eliminates the need to find additional sources of tumor masks to provide as conditions during the inference phase of the LDM, which is impractical in multi-site study settings. Moreover, acquiring the tumor mask along with the four MRI contrasts is essential, as we require the tumor masks with IDH mutation status to train IDH classification models. By generating the masks within the LDM, we eliminate the need for additional segmentation tools to delineate the whole tumor prior to training the classification models, thereby streamlining the overall workflow.

## Autoencoder

We modified the 2D autoencoder architecture to accommodate 3D multi-contrast MRI inputs, effectively capturing spatial dependencies across all three dimensions. To eliminate the need to provide any tumor mask condition or use separate tumor segmentation models prior to training our classification model, the input to the autoencoder during training consists of four MRI sequences and the whole tumor mask. By including the tumor mask as part of the training input, the model learns to generate both the tumor segmentation and the corresponding MRI sequences simultaneously during inference.

The autoencoder model was specifically developed to process high-dimensional multi-contrast MRI data. It includes a variational autoencoder (VAE)[45] with a super-resolution (SR) module within the decoder, enabling efficient encoding, reconstruction, and spatial resolution enhancement of input images. The VAE architecture employs residual blocks as the fundamental units for both the encoder and decoder, with a vanilla attention module included in the attention layers to improve feature representation. Several loss functions were utilized to optimize the VAE model, including pixel-wise loss, perceptual loss[46], Kullback-Leibler (KL) divergence loss[47], and discriminator loss. Additional loss components, such as the structural similarity index (SSIM)[48] and gradient losses, were also evaluated but did not significantly enhance image quality.

To further enhance the spatial resolution of the reconstructed images, the decoder was augmented with an SR module based on a

standard UNet architecture. The model was trained in two stages: first, the VAE was trained without the SR module, and then the decoder was fine-tuned with the SR module using the pre-trained VAE model. The SR module training leveraged perceptual loss, pixel-wise loss, and SSIM loss to ensure high-quality reconstructions.

The autoencoder model takes inputs and outputs with dimensions of $128 \times 192 \times 192$ (depth × height × width) and five channels, including four MRI sequences and the whole tumor mask. The latent space is represented by a compact dimensionality of $8 \times 16 \times 32 \times 32$ (channels × depth × height × width). The model was trained on four A100 GPUs using a distributed data-parallel strategy. A batch size of 1 and a 16-bit precision format were adopted to accommodate the computational demands of high-dimensional data. Details of the network architecture and training hyperparameters are presented in Table S2.

## Diffusion model

The diffusion model was also adapted from a 2D to a 3D architecture to fully leverage the volumetric nature of MRI data. We incorporated cross-attention layers to condition the model on text input, specifically the IDH mutation status or the tumor types. This conditioning mechanism enables the model to generate MRI data consistent with specific conditions. As the complexity of 3D MRI data increases compared to 2D images, we enhanced the model's capacity by increasing the number of down-sampling levels in the denoising UNet and the UNet base channel size. These modifications allow the model to learn more intricate features from the MRI data, ensuring that the synthetic images are realistic. Supplementary Note 3 provides details on the LDM's configuration and training hyperparameters.

We leveraged the LDM as an alternative to FL by generating synthetic data for centralized training. Instead of using an intensive back-and-forth to share and update model parameters like FL, LDMs were used to generate synthetic MRI data. Each center trained an LDM using local MRI data to capture the distribution of the underlying data. The trained LDMs were transmitted to a central server to generate site-specific synthetic MRI samples for each center. These synthetic samples were combined to train centralized classification models, allowing the model to learn from aggregated data characteristics while maintaining data privacy. Supplementary Note 7 provides a comparison of the computational and communication overhead associated with centralized training, FL, and CATphishing.

To assess the effectiveness of CATphishing, we initially generated 1600 IDH-mutated and 1600 IDH-wildtype samples for each participating site. To enhance diversity and eliminate redundancy among synthetic samples, we implemented a filtering process. Specifically, we extracted radiomics features from the whole tumor[49] and calculated the distances between the radiomics features for all sample pairs within each site. Samples exhibiting high similarity, those with distances below the 20th percentile of the distance distribution among synthetic samples at each site, were excluded. This filtering process resulted in combined training sample sizes of 2087 mutated and 2527 wild-type samples across the three sites.

## Statistical analysis

We used the Fréchet Inception Distance (FID)[29] to evaluate the fidelity of synthetic samples. The quality of the synthetic samples was also assessed using no-reference image quality assessment metrics, including the Blind/Referenceless Image Spatial Quality Evaluator (Brisque)[50] and Perception-based Image Quality Evaluator (PIQE)[32]. For the IDH classification models, we used standard classification metrics, including accuracy (ACC), sensitivity (SEN), specificity (SPE), and receiver operating characteristic area under the curve (AUC), to evaluate their performance. The AUC confidence interval (CI) was estimated using the bootstrap distribution. Comparative analyses were

**Table 5 | Summary of the tumor-type classification performance of different models trained on real or synthetic MRI data**

| Stage | Experiments | Training data | Metrics | UCSF | EGD | UWM | UTSWp2 | Overall | Model Comparison |
|---|---|---|---|---|---|---|---|---|---|
| Stage 1: (Oligodendroglioma + Astrocytoma) vs. Glioblastoma | Centralized Training | Real TCGA + Real UTSW + Real NYU | ACC | 96.5 | 97.3 | 95.8 | 96.6 | 96.7 | - |
| | | | SEN | 91.4 | 96.6 | 86.7 | 97.0 | 94.4 | |
| | | | SPE | 97.7 | 97.7 | 96.5 | 96.5 | 97.3 | |
| | | | AUC | 0.982 | 0.992 | 0.977 | 0.997 | 0.988 | |
| | FL using the FedAvg algorithm with 100 FL rounds | Real TCGA + Real UTSW + Real NYU | ACC | 96.1 | 97.6 | 94.9 | 96.6 | 96.5 | $\chi^2(1) = 0.211$, $p = 0.6464$ |
| | | | SEN | 88.2 | 97.2 | 86.7 | 93.9 | 93.4 | |
| | | | SPE | 98.0 | 97.7 | 95.5 | 97.4 | 97.3 | |
| | | | AUC | 0.986 | 0.992 | 0.978 | 0.997 | 0.990 | |
| | Centralized Training using Synthetic Samples | Synthetic TCGA + Synthetic UTSW + Synthetic NYU | ACC | 95.7 | 95.6 | 96.3 | 95.2 | 95.7 | $\chi^2(1) = 0.0976$, $p = 0.7548$ |
| | | | SEN | 88.2 | 95.2 | 93.3 | 93.9 | 92.7 | |
| | | | SPE | 97.5 | 95.8 | 96.5 | 95.6 | 96.5 | |
| | | | AUC | 0.977 | 0.981 | 0.974 | 0.993 | 0.980 | |
| Stage 2: Oligodendroglioma vs. Astrocytoma | Centralized Training | Real TCGA + Real UTSW + Real NYU | ACC | 73.1 | 80.7 | 73.3 | 66.7 | 76.2 | - |
| | | | SEN | 66.7 | 88.7 | 50.0 | 62.5 | 81.6 | |
| | | | SPE | 73.8 | 73.0 | 76.9 | 70.6 | 73.4 | |
| | | | AUC | 0.739 | 0.874 | 0.577 | 0.820 | 0.844 | |
| | FL using the FedAvg algorithm with 100 FL rounds | Real TCGA + Real UTSW + Real NYU | ACC | 67.7 | 81.4 | 80.0 | 66.7 | 75.2 | $\chi^2(1) = 0.593$, $p = 0.4414$ |
| | | | SEN | 66.7 | 84.5 | 50.0 | 56.3 | 77.6 | |
| | | | SPE | 67.9 | 78.4 | 84.6 | 76.5 | 73.9 | |
| | | | AUC | 0.729 | 0.895 | 0.654 | 0.779 | 0.839 | |
| | Centralized Training using Synthetic Samples | Synthetic TCGA + Synthetic UTSW + Synthetic NYU | ACC | 68.8 | 80.7 | 73.3 | 69.7 | 75.2 | $\chi^2(1) = 0.432$, $p = 0.5108$ |
| | | | SEN | 66.7 | 83.1 | 50.0 | 62.5 | 77.6 | |
| | | | SPE | 69.1 | 78.4 | 76.9 | 76.5 | 73.9 | |
| | | | AUC | 0.748 | 0.862 | 0.539 | 0.787 | 0.821 | |
| Three-class classification: Oligodendroglioma vs. Astrocytoma vs. Glioblastoma | Centralized Training | Real TCGA + Real UTSW + Real NYU | ACC | 92.2 | 91.4 | 94.4 | 89.1 | 91.9 | - |
| | | | ACC - Oligo | 66.7 | 87.3 | 50.0 | 62.5 | 80.6 | |
| | | | ACC - Astro | 70.2 | 68.9 | 69.2 | 70.6 | 69.7 | |
| | | | ACC - GBM | 97.5 | 97.7 | 96.5 | 95.6 | 97.1 | |
| | FL using the FedAvg algorithm with 100 FL rounds | Real TCGA + Real UTSW + Real NYU | ACC | 91.1 | 91.8 | 93.5 | 89.1 | 91.5 | $\chi^2(1) = 0.195$, $p = 0.9071$ |
| | | | ACC - Oligo | 66.7 | 83.1 | 50.0 | 56.3 | 76.5 | |
| | | | ACC - Astro | 63.1 | 75.7 | 69.2 | 76.5 | 69.7 | |
| | | | ACC - GBM | 97.7 | 97.7 | 95.5 | 95.6 | 97.0 | |
| | Centralized Training using Synthetic Samples | Synthetic TCGA + Synthetic UTSW + Synthetic NYU | ACC | 91.1 | 90.0 | 94.4 | 87.8 | 90.9 | $\chi^2(1) = 0.882$, $p = 0.6434$ |
| | | | ACC - Oligo | 66.7 | 80.3 | 50.0 | 62.5 | 75.5 | |
| | | | ACC - Astro | 65.5 | 77.0 | 69.2 | 76.5 | 71.3 | |
| | | | ACC - GBM | 97.2 | 95.4 | 96.5 | 93.0 | 96.1 | |

Statistical comparisons were performed using the McNemar test for stage 1 and stage 2 (binary classifications) and the Stuart-Maxwell test for the final 3-class classification All tests are two-sided, and no adjustments are applied. Source data are provided as a Source Data file.

conducted among centralized training with shared data, FL, and the diffusion-based approach. Additionally, McNemar's test[51] was applied to assess significant differences in the performance of the IDH classification models, while the Stuart-Maxwell test[52] was employed to evaluate differences in the three-class tumor-type classification results.

### Reporting summary

Further information on research design is available in the Nature Portfolio Reporting Summary linked to this article.

## Data availability

This study utilized retrospective MRI scans from four publicly available and three internal datasets. Public datasets: The TCGA dataset is available at https://www.cancer.gov/ccg/research/genome-sequencing/tcga. The EGD dataset is available at https://xnat.bmia.nl/data/archive/projects/egd The University of California San Francisco Preoperative Diffuse Glioma MRI dataset is available at https://www.cancerimagingarchive.net/collection/ucsf-pdgm/ The University of

Pennsylvania glioblastoma cohort is available at https://www.cancerimagingarchive.net/collection/upenn-gbm/ Internal datasets: The internal datasets obtained from UT Southwestern Medical Center (UTSW) are planned for public release on The Cancer Imaging Archive (TCIA) pending acceptance of a separate dataset-focused manuscript that is currently under peer review. Once approved and deposited, the dataset will be accessible through TCIA without cost to the research community. Access will require user registration and agreement to TCIA's data usage policies. A permanent Digital Object Identifier (DOI) and direct hyperlink to the dataset will be provided in the manuscript at the time of release. We anticipate that the dataset will become publicly available within 6-12 months following publication of this study, depending on the timing of acceptance and curation by TCIA. - The internal datasets from New York University (NYU) and the University of Wisconsin-Madison (UWM) cannot be shared publicly because of institutional and patient privacy regulations. Trained latent diffusion models to generate synthetic data used in this study will be shared under a data use agreement for research purposes. Requests can be initiated by

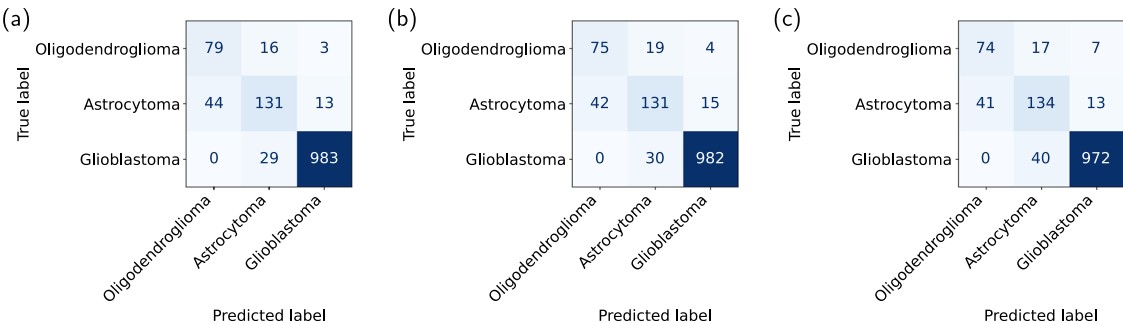

**Fig. 4 | Confusion matrices for tumor-type classification using three training approaches. a** Centralized training using real data, **b** FL using real data, **c** CATphishing. Each matrix displays the classification performance across three tumor types: oligodendroglioma, astrocytoma, and glioblastoma. Source data are provided as a Source Data file.

contacting the corresponding author. Source data are provided with this paper.

## Code availability

All source code, configuration files, and training scripts used in this study are available in the Code Ocean capsule with https://doi.org/10.24433/CO.4976589.v1.

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

## Acknowledgements

This research was supported by the NIH/NCI R01CA260705 (J.A.M.).

## Author contributions
Guarantors of integrity of entire study, N.C.D.T., C.G.B.Y., M.C.P., J.A.M.; study concepts/study design, N.C.D.T., C.G.B.Y., M.C.P., J.A.M.; data acquisition, B.C.W., J.M.H., D.D.R., N.S., J.B., K.J.H., M.D.L., R.J., R.J.B.; manuscript drafting or manuscript revision for important intellectual content, N.C.D.T., C.G.B.Y., J.M.H., M.D.L., R.J., B.F., A.J.M., M.C.P., J.A.M.; approval of final version of submitted manuscript, all authors; agrees to ensure any questions related to the work are appropriately resolved, all authors; literature research, N.C.D.T., C.G.B.Y., M.C.P., J.A.M.; clinical studies, N.S., K.J.H., T.R.P., M.D.L., R.J., R.J.B., M.C.P.; experimental studies, N.C.D.T., C.G.B.Y., J.A.M.; statistical analysis, N.C.D.T., N.S., K.J.H., R.J.B.; and manuscript editing, N.C.D.T., C.G.B.Y., B.C.W., J.M.H., D.D.R., N.S., J.B., K.J.H., T.R.P., B.F., M.D.L., R.J., R.J.B., A.J.M., M.C.P., J.A.M.

## Competing interests
The authors declare no competing interests.
