## [Peer Review File · Nature Communications]

Categorical and Phenotypic Image Synthetic Learning as an Alternative to Federated Learning

Corresponding Author: Dr Nghi Truong

Version 0:

Reviewer comments:

Reviewer #1

(Remarks to the Author)

Authors introduce the CATphishing approach by training local LDMs and generating synthetic samples at a central site. The approach is tested using a classification task regarding IDH mutations.

Strong points:

- Using extensive data samples from different sites to support author claims
- Systematic comparison between central, federated and the LDM approach showing en par performances.
- The methodological approach could be of high relevance, but the novelty is unclear

Weaknesses:

- I am not sure if no prior work has leveraged LDMs for 3D MRI data as the authors claim. Generation of synthetic samples using LDM is not new, also in the 3D MRI domain (e.g. <https://arxiv.org/abs/2411.01351>). Maybe the authors could provide more clarification on this the uniqueness of their work.
- Central storage of synthetic data – derived from original patient data – may still hold privacy issues even more than the FL approach where only model parameters are processed on a central server.
- The example task of IDH mutation is known to be very simple (resulting generally in high AUCs) and clinically less relevant, as IDH mutation analysis is not expensive, so why using MRI analyses which only reproduces a cost-effective genetic marker with less accuracy. The high AUCs provide less discrimination when comparing the three approaches. Evaluations on further medical tasks are therefore required
- In their discussion and as main rationale, authors describe, that FL adds „significant overhead due to the continuous back-and-forth exchanges requiring simultaneous training at all“. However, FL does not need to be applied synchronously at all sites. More clarification on this would be helpful.

(Remarks on code availability)

Reviewer #2

(Remarks to the Author)

This article introduces a new method called CATphishing (Categorical and Phenotypic Image Synthetic Learning), which aims to address privacy, communication, and data-sharing issues in multi-center collaborative development of machine learning models for medical imaging. This approach holds certain clinical value.

I have some questions regarding this work:

1. The synthetic model presented in the paper focuses on the most fundamental sequences, including T1, T2, T2 FLAIR, and T1_C. However, current multi-center radiomics research often extends beyond structural imaging analysis and places greater emphasis on functional MRI (e.g., DTI, resting-state MRI) or metabolic MRI (e.g., ASL, CEST). Can the model used in the study also be applied to these more advanced imaging modalities?
2. I tend to describe the IDH mutation classification in the paper as a classification of whether there is an enhancement lesion on T1 contrast-enhanced MRI. I am unsure whether astrocytoma (WHO Grade 4, IDH mutant), glioma of IDH wild-type, WHO Grade 4 (histological Grade 2 with non-enhanced region after contrast) would still exhibit strong synthetic effects and IDH prediction accuracy. I suggest the authors provide visual representations of these images in the appropriate sections in the manuscript.

3. In the testing cohort, the proportion of IDH-mutant to wild-type cases (180/1133) shows a significant imbalance. Could this lead to bias in the final validation results? I recommend adjusting the case ratio within the testing cohort.

4. In Figure 2, the displayed MR slides are all from the "lateral ventricle level", where MRI has relatively higher SNR. However, for slices near the skull base or large arteries (e.g., vertebral artery and internal carotid artery), imaging artifacts caused by bone structures and arterial pulsation could potentially affect synthetic imaging. Additionally, the lesions shown in the figure are all larger than 4 cm in diameter. Can this method achieve equally effective results for small enhancing lesions, diffuse midline gliomas, or tumors near large veins (e.g., venous sinuses, the Galen vein, including pineal region tumors)?

5. The IDH mutation prediction task in the paper appears relatively simple for Nature Communications. I suggest incorporating additional molecular pathology prediction tasks for gliomas, as such data are readily available in TCGA.

(Remarks on code availability)

Reviewer #3

(Remarks to the Author)

This paper compares latent diffusion models (LDMs) against other common methods for multi-site collaborations, namely, traditional centralized training and federated learning. They were able to achieve comparable results with LDMs as with the other methods with several notable advantages. These include, but are not limited to, decreased burden of interactive parameter sharing in federated learning and increased data privacy compared to traditional centralized training, which at times pose significant barriers in multi-site collaborations in medical imaging. In terms of methodology, the evaluation of synthetic data and the comparison of results among the three methods is sound. In particular, the authors' method of synthetic tumor mask generation along with synthetic images is an elegant application of LDMs.

One area of relative weakness is their discussion on membership inference attacks (MIAs). However, the authors do address safeguards taken within their own methods as well as limitations, and further discussion on this may be beyond the scope of this study as MIAs are a vulnerability shared by all the methods discussed. It would further strengthen the paper to include a comparison of computational costs among the three methods to highlight the potential increase in efficiency with their proposed method.

Overall, though this method does not necessarily provide superior results compared to other methods, the authors provide compelling data that show the suitability and advantages of leveraging LDMs in multi-site collaborations.

(Remarks on code availability)

This code provides adequate organization and documentation to allow for reproducibility

Version 1:

Reviewer comments:

Reviewer #1

(Remarks to the Author)

The authors have addressed all of my issues. They have added analyses regarding data privacy and one classification task and provided valuable results.

(Remarks on code availability)

Reviewer #2

(Remarks to the Author)

Thank you to the author team for the response. All my main questions have now been answered, and I recommend that the journal accept this article.

(Remarks on code availability)

Reviewer #3

(Remarks to the Author)

The authors have addressed all significant concerns that were raised with the initial review. Though there remains limitations in translation to clinically relevant practice due to the task chosen for the machine learning task, this does not necessarily detract from the main purpose of comparing LDMs to federated and centralized training methods. As such, the work presented in this paper ultimately has the potential to lay the groundwork for future work that may address different,

more clinically relevant machine learning tasks, and I have no major reservations in regards to publication.

(Remarks on code availability)

Dear Reviewers,

We would like to sincerely thank the editors and reviewers for their diligent review and valuable feedback. We have carefully considered their comments and incorporated the necessary modifications in the manuscript, which are listed below point by point.

REVIEWER COMMENTS

Reviewer expertise:

Reviewer #1: AI and health informatics

Reviewer #2: Brain cancer radiology, AI

Reviewer #3: Brain cancer radiology, AI, federated learning

Reviewer #1 (Remarks to the Author):

Authors introduce the CATphishing approach by training local LDMs and generating synthetic samples at a central site. The approach is tested using a classification task regarding IDH mutations.

Strong points:

- Using extensive data samples from different sites to support author claims
- Systematic comparison between central, federated and the LDM approach showing en par performances.
- The methodological approach could be of high relevance, but the novelty is unclear

Weaknesses:

- I am not sure if no prior work has leveraged LDMs for 3D MRI data as the authors claim. Generation of synthetic samples using LDM is not new, also in the 3D MRI domain (e.g. <https://arxiv.org/abs/2411.01351>). Maybe the authors could provide more clarification on this the uniqueness of their work.

We thank the reviewer for the valuable comment and the reference to relevant literature. We agree that the generation of synthetic 3D MRI samples using LDMs has been extensively explored recently. However, our intention was not to claim the originality in the application of LDMs to 3D MRI generation, but rather to highlight a novel use of LDM-generated synthetic data as an alternative to Federated Learning (FL) for multi-center collaboration. While some works have integrated DMs into FL pipelines, they still rely on a federated communication framework. In contrast, our CATphishing approach utilizes synthetic data generation via LDMs to fully eliminate the need for cross-site model or real data exchange, which, to the best of our knowledge, has not been previously explored in the medical imaging domain, particularly for 3D MRI. We have revised the 3rd paragraph of the Introduction section to clarify this.

- Central storage of synthetic data – derived from original patient data – may still hold privacy issues even more than the FL approach where only model parameters are processed on a central server.

We thank the reviewer for highlighting potential privacy risks associated with the central storage of synthetic data. To evaluate this concern, we performed a membership inference attack (MIA) using a distance-based method with the Structural Similarity Index Measure (SSIM), as detailed in Section D of the Supplemental Materials. We tested whether real training images (“members”) could be distinguished from held-out samples (“non-members”) based on their maximum SSIM similarity to the synthetic images. The results showed chance-level performance (AUC = 0.493) and substantial overlap in SSIM distributions between members and non-members (Figure S1, Supplemental Materials), indicating no meaningful distinction. These findings suggest that our synthetic data does not retain identifiable features from training samples and poses minimal membership-related privacy risk.

We agree that privacy continues to be a critical concern and that membership inference represents just one aspect of potential leakage. We have revised the 5th paragraph of the Discussion section and will pursue further evaluations using more robust MIA techniques.

- The example task of IDH mutation is known to be very simple (resulting generally in high AUCs) and clinically less relevant, as IDH mutation analysis is not expensive, so why using MRI analyses which only reproduces a cost-effective genetic marker with less accuracy. The high AUCs provide less discrimination when comparing the three approaches. Evaluations on further medical tasks are therefore required

We appreciate the reviewer’s insightful comment regarding the simplicity of the IDH mutation classification task. To demonstrate the broader applicability and clinical relevance of CATphishing, we incorporated a second downstream task, tumor-type classification based on the 2021 WHO criteria, which integrates both IDH and 1p19q status to distinguish oligodendroglioma, astrocytoma, and glioblastoma. We adopted a two-stage classification pipeline:

- *Stage 1 distinguishes IDH-mutated and wild-type cases, with the latter classified as glioblastoma.*
- *Stage 2 further classifies the IDH-mutant subset into oligodendroglioma versus astrocytoma.*

Additionally, we included a separate cohort from UTSW as a new testing set. As shown in the newly added results (Table 5), models trained on synthetic data via CATphishing achieved comparable performance to those trained using centralized and federated approaches on real

data. This expanded evaluation highlights the capability of CATphishing to support complex multi-class classification tasks beyond IDH status alone. We have revised the manuscript throughout to incorporate this new experiment.

- In their discussion and as main rationale, authors describe, that FL adds "significant overhead due to the continuous back-and-forth exchanges requiring simultaneous training at all". However, FL does not need to be applied synchronously at all sites. More clarification on this would be helpful.

We appreciate the reviewer's comment regarding the flexibility of FL protocols. FL can indeed be implemented in both synchronous and asynchronous modes. Our study employed a synchronous FL framework, and our discussion specifically referred to this mode, where all participating sites must complete local training and upload model weights before the server performs aggregation. We agree that asynchronous FL can mitigate some of these constraints by allowing participating sites to upload updates independently, without waiting for others. However, even in an asynchronous setting, FL still involves repeated bidirectional communication between the participating sites and the central server across multiple rounds. Moreover, it introduces additional complexities, such as tracking client versions, handling stale updates, or addressing fairness concerns.

To clarify this point, we have revised the 2nd paragraph of the discussion section to acknowledge asynchronous FL and further discuss its potential advantages and limitations.

Reviewer #2 (Remarks to the Author):

This article introduces a new method called CATphishing (Categorical and Phenotypic Image Synthetic Learning), which aims to address privacy, communication, and data-sharing issues in multi-center collaborative development of machine learning models for medical imaging. This approach holds certain clinical value.

I have some questions regarding this work:

1. The synthetic model presented in the paper focuses on the most fundamental sequences, including T1, T2, T2 FLAIR, and T1_C. However, current multi-center radiomics research often extends beyond structural imaging analysis and places greater emphasis on functional MRI (e.g., DTI, resting-state MRI) or metabolic MRI (e.g., ASL, CEST). Can the model used in the study also be applied to these more advanced imaging modalities?

We thank the reviewer for this comment. Our current study focuses on the most widely available and routinely acquired structural MRI sequences, T1, T1 post contrast, T2, and FLAIR, because they serve as the foundational inputs for nearly all clinical brain tumor workflows and are

consistently present across multi-center datasets. These sequences also form the basis for many radiomics and deep learning studies, enabling broader applicability and reproducibility.

We agree that emerging research increasingly incorporates functional MRI (e.g., diffusion tensor imaging, resting-state fMRI) and metabolic MRI (e.g., arterial spin labeling, chemical exchange saturation transfer) for richer tumor characterization. The latent diffusion model (LDM) framework we employ is modality-agnostic in its design and can, in principle, be extended to these advanced imaging types, provided there is sufficient high-quality, co-registered training data. We have revised the last paragraph of the Discussion section to clarify this point and outline it as a promising direction for future development.

2. I tend to describe the IDH mutation classification in the paper as a classification of whether there is an enhancement lesion on T1 contrast-enhanced MRI. I am unsure whether astrocytoma (WHO Grade 4, IDH mutant), glioma of IDH wild-type, WHO Grade 4 (histological Grade 2 with non-enhanced region after contrast) would still exhibit strong synthetic effects and IDH prediction accuracy. I suggest the authors provide visual representations of these images in the appropriate sections in the manuscript.

We thank the reviewer for this insightful comment. We agree that visual inspection of contrast enhancement patterns, especially on T1 post-contrast images, may sometimes correlate with molecular subtypes, such as IDH status. However, the relationship is not absolute. For example, some IDH-wildtype gliomas may lack enhancement, while certain IDH-mutated gliomas may exhibit enhancement. To address the reviewer's concern and to clarify the model's behavior across different subtypes, we have included additional visual examples of synthetic MRIs in Supplementary Figures S2-S4. These examples represent three tumor types: oligodendroglioma (IDH-mutated, 1p19q co-deleted), astrocytoma (IDH-mutated, 1p19q non-codeleted), and glioblastoma (IDH-wildtype). These figures demonstrate the diversity of imaging phenotypes captured by synthetic data, including both enhancing and non-enhancing lesions. This supports the model's ability to generalize across imaging presentations and suggests that the classifier's performance is not solely driven by enhancement cues.

3. In the testing cohort, the proportion of IDH-mutant to wild-type cases (180/1133) shows a significant imbalance. Could this lead to bias in the final validation results? I recommend adjusting the case ratio within the testing cohort.

We appreciate the reviewer's comment regarding the class imbalance in the IDH testing cohort. To address potential bias and ensure a more comprehensive validation, we included a new held-out cohort from UTSW (UTSWp2), resulting in an updated testing set comprised of 230 mutated and 1247 wildtype cases. Moreover, we conducted an additional evaluation of the IDH mutation

classification task as part of the two-stage tumor-type classification framework. Specifically, the first stage of the tumor-type classification framework classifies the IDH-mutated versus IDH-wildtype. We adopted a different training/testing split for the tumor-type classification experiment, using TCGA, UTSW, and NYU (200 mutated vs. 491 wildtype) for training the first stage. We conducted a comprehensive evaluation of the first stage of IDH classification using five independent datasets: EGD, UCSF, UWM, UPenn, and the new held-out cohort from UTSW. This results in a testing cohort of 332 mutated and 1,420 wildtype cases, which provides a more rigorous evaluation.

The results of this extended evaluation are presented in Table S4 of Section F in the Supplementary Materials, showing consistent performance and demonstrating that the classifier maintains generalizability even across more diverse sites. This expanded evaluation supports the validity and stability of our approach across diverse testing scenarios.

4. In Figure 2, the displayed MR slides are all from the “lateral ventricle level”, where MRI has relatively higher SNR. However, for slices near the skull base or large arteries (e.g., vertebral artery and internal carotid artery), imaging artifacts caused by bone structures and arterial pulsation could potentially affect synthetic imaging. Additionally, the lesions shown in the figure are all larger than 4 cm in diameter. Can this method achieve equally effective results for small enhancing lesions, diffuse midline gliomas, or tumors near large veins (e.g., venous sinuses, the Galen vein, including pineal region tumors)?

We thank the reviewer for this thoughtful comment regarding variations in image quality and lesion characteristics across anatomical regions. To address this concern, we have included additional examples in Supplemental Figures S2-S4, which specifically showcase synthetic MRI samples from various brain regions and tumor volumes. These examples demonstrate the generative model’s capability to produce realistic anatomical and pathological features. We have also revised the manuscript accordingly to incorporate these new samples.

5. The IDH mutation prediction task in the paper appears relatively simple for Nature Communications. I suggest incorporating additional molecular pathology prediction tasks for gliomas, as such data are readily available in TCGA.

We thank the reviewer for the thoughtful comment regarding the relative simplicity of the IDH mutation classification task. To demonstrate the broader utility and clinical relevance of CATphishing, we introduced an additional downstream task: tumor-type classification following the 2021 WHO guidelines, which incorporate both IDH and 1p19q status to differentiate oligodendroglioma, astrocytoma, and glioblastoma.

We implemented a two-step classification approach:

- *The first stage distinguishes between IDH-mutated and IDH-wildtype tumors, with the latter categorized as glioblastoma.*
- *The second stage classifies IDH-mutant cases as either oligodendroglioma or astrocytoma.*

To further validate model generalizability, we included an independent test cohort from UTSW. As presented in the newly added Table 5 and Figure 4, models trained on synthetic data using CATphishing performed comparably to those trained on real data via centralized and federated learning. These results underscore CATphishing's effectiveness in handling more complex, clinically relevant multi-class classification tasks. The manuscript has been updated accordingly to reflect this expanded analysis.

Reviewer #3 (Remarks to the Author):

This paper compares latent diffusion models (LDMs) against other common methods for multi-site collaborations, namely, traditional centralized training and federated learning. They were able to achieve comparable results with LDMs as with the other methods with several notable advantages. These include, but are not limited to, decreased burden of interactive parameter sharing in federated learning and increased data privacy compared to traditional centralized training, which at times pose significant barriers in multi-site collaborations in medical imaging. In terms of methodology, the evaluation of synthetic data and the comparison of results among the three methods is sound. In particular, the authors' method of synthetic tumor mask generation along with synthetic images is an elegant application of LDMs.

One area of relative weakness is their discussion on membership inference attacks (MIAs). However, the authors do address safeguards taken within their own methods as well as limitations, and further discussion on this may be beyond the scope of this study as MIAs are a vulnerability shared by all the methods discussed. It would further strengthen the paper to include a comparison of computational costs among the three methods to highlight the potential increase in efficiency with their proposed method.

We appreciate the reviewer's suggestion to include a comparison of computational costs, as this is an important consideration for the deployment of the frameworks. We have now incorporated Table S5 in the Supplemental Materials, providing a comparison of the computational and communication overhead associated with each method. Specifically, centralized learning requires the transfer of private imaging data to a central server, which poses substantial privacy risks. However, it allows for standard, single-site training without coordination overhead. FL, while avoiding data centralization, involves repeated cycles of model training, weight transfer, and aggregation across all participating sites over multiple communication rounds (e.g., 100

rounds in our study). This introduces significant cumulative communication overhead, synchronization complexity, and energy costs. Finally, CATphishing involves a one-time local training of a latent diffusion model (LDM) per site, followed by a single model upload to a central location for synthetic data generation and downstream training. This drastically reduces communication and coordination requirements.

Overall, though this method does not necessarily provide superior results compared to other methods, the authors provide compelling data that show the suitability and advantages of leveraging LDMs in multi-site collaborations.

Reviewer #3 (Remarks on code availability):

This code provides adequate organization and documentation to allow for reproducibility

We believe these revisions have addressed the concerns raised during the review process and have strengthened the overall quality and clarity of the manuscript. We sincerely hope that the manuscript is now suitable for publication in Nature Communications.

Thank you once again for the invaluable feedback and for considering our work.

Sincerely,

Nghi C. D. Truong, PhD